# Design and Application of Multi-Dimensional Visualization System for Large-Scale Ocean Data

Teng Lv , Jun Fu and Bao Li *

Department of Navigation Engineering, Naval University of Engineering, Wuhan 430000, China
* Correspondence: 13545299580@163.com

**Abstract:** With the constant deepening of research on marine environment simulation and information expression, there are higher and higher requirements for the sense of the reality of ocean data visualization results and the real-time interaction in the visualization process. Aiming at the challenges of 3D interactive key technology and GPU-based visualization algorithm technology, we developed a visualization system for large-scale 3D marine environmental data. The system realizes submarine terrain rendering, contour line visualization, isosurface visualization, section visualization, volume visualization and flow field visualization. In order to manage and express the data in the system, we developed a data management module, which can effectively integrate a large number of marine environmental data and manage them effectively. We developed a series of data analysis functions for the system, such as point query and line query, local analysis and multi-screen collaboration, etc. These functions can effectively improve the data analysis efficiency of users and meet the data analysis needs in multiple scenarios. The marine environmental data visualization system developed in this paper can efficiently and intuitively simulate and display the nature and changing process of marine water environmental factors.

**Keywords:** marine environment; 3D data; OSG; NetCDF; multi-dimensional visualization

## 1. Introduction

The 21st century is the century of the ocean, and the relationship between the production and operation of society and the ocean is getting closer and closer. With the rapid development of ocean exploration methods, how to process and analyze huge ocean data has become an urgent problem to be solved. At the same time, information visualization is becoming the mainstream means of data analysis, and its intuitive and clear characteristics can significantly improve the efficiency of data analysis. At present, in various disciplines, visualization systems have become a common research tool for data processing and analysis.

Marine data have the characteristics of being multi-dimensional and complex, and it is difficult to express its characteristics intuitively and comprehensively using traditional data analysis methods. Therefore, more and more people choose to use 3D visualization to process and analyze marine data [1]. The 3D visualization of the ocean has the characteristics of intuitiveness, time sharing and regularity, which can intuitively convey the internal relationship and time series characteristics of ocean data. The marine visualization system can integrate the advantages of visualization technology and develop targeted functions according to the needs of users, which can convey and perform marine scenes in an intuitive way [2]. However, this is not an easy job. How to effectively integrate massive ocean data and organically integrate it into the visualization scene as required and give the system an interactive user interface is the challenge of the current visualization system development.

The visualization research of marine environmental data is mainly reflected in the simulation and analysis of marine environmental elements such as temperature, salinity,

density, speed of sound, and ocean currents. Many institutions and scientific researchers have carried out a series of targeted development around the integrated management and three-dimensional visualization of diverse, multi-dimensional and dynamic marine data. For example, software platforms such as *Google Ocean* developed by Google, *Skyline* developed by Skyline, *MyOcean* developed by the European Union, *ArcGlobe* developed by ESRI, *WorldWind* [3] developed by NASA and other software platforms provide a spatial information integration interface for building virtual earth models and perform on top of it; GIS environment-based software systems such as Marine GIS developed by the University of New Hampshire and ArcGIS developed by ESRI provide the application foundation and operability for marine environmental data visualization; in terms of environmental data visualization, Ravi et al. [4] investigated and summarized its technology and tools used. Emmanuelle et al. [5] conducted extensive research on computer graphics used in ocean simulation and rendering. Gao Xizhang et al. [6] selected OpenGL and IDL as the 3D visualization development technology based on the GIS environment and developed the 3D visualization component CMa3DView using the C++ and VC++ languages, then integrated it into visualized operation in the self-developed Maxplorer software; Zhang Feng et al. [7] built a virtual earth model based on the Skyline platform, on which the data of the DEM (Digital Elevation Model), basic geographic remote sensing data and texture data were integrated and visualized; based on the OSG 3D rendering engine, Li Xinfang et al. [8] established a 3D Virtual Earth model with VC++ language and processed the existing DEM data and satellite cloud images on the model to build a real-time, efficient and realistic 3D simulation and visualization system of a marine environment; Chen Ge et al. [9,10] designed a 3D ocean virtual visualization engine named VV-Ocean to achieve high-fidelity simulation of the ocean environment, visualization of ocean multi-dimensional data and reproduction of ocean life, including virtual ocean scene display, ocean environment data dynamic real-time visualization and an intuitive simulation of marine life and other functions; Sun Qiang et al. [11] designed a marine scalar field visualization system based on the open-source 3D earth framework Cesium, realized the visualization and analysis of marine environment elements based on the B/S architecture and systematically explained the visualization system architecture method.

However, the above research is either limited to the visualization of marine environmental elements without adding necessary data analysis functions, or only develops and researches some visualization algorithms without integrating multiple visualization methods into an organic system. Aiming at this research blind spot, this paper tries to design a marine data visualization system that can adapt to a variety of scientific research scenarios. We believe that an ocean visualization system that meets the needs of scientific research should have the following characteristics:

(1) Ability to integrate and manage large amounts of marine data.
(2) Achieve efficient real-time rendering.
(3) Multi-functional and interactive system interface.

Taking this as the research background, this paper considers the characteristics of large-scale marine environment data and the use characteristics of the visualization system and develops a visualization system based on GPU parallel rendering technology and NetCDF data storage technology through point, line, surface, and volume model analysis. It mainly includes a variety of visualization algorithms such as contour tracing, isosurface generation, section rendering, volume rendering etc., and realizes the three-dimensional visualization of large-scale marine hydrological environment data. On this basis, more importantly, different from ordinary marine environment data visualization software, the application scenario for this system is mainly the underwater navigation of submersibles. Therefore, our main concern is the stability, compatibility of the visualization algorithm and integrity of the necessary functions of the system. Taking this as the design logic, we extensively solicited the usage needs of the user group and designed the visualization system of this article.

## 2. System Framework

The system structure designed in this paper is mainly composed of a data processing part, a data rendering part and an application display part. The specific structure is shown in Figure 1.

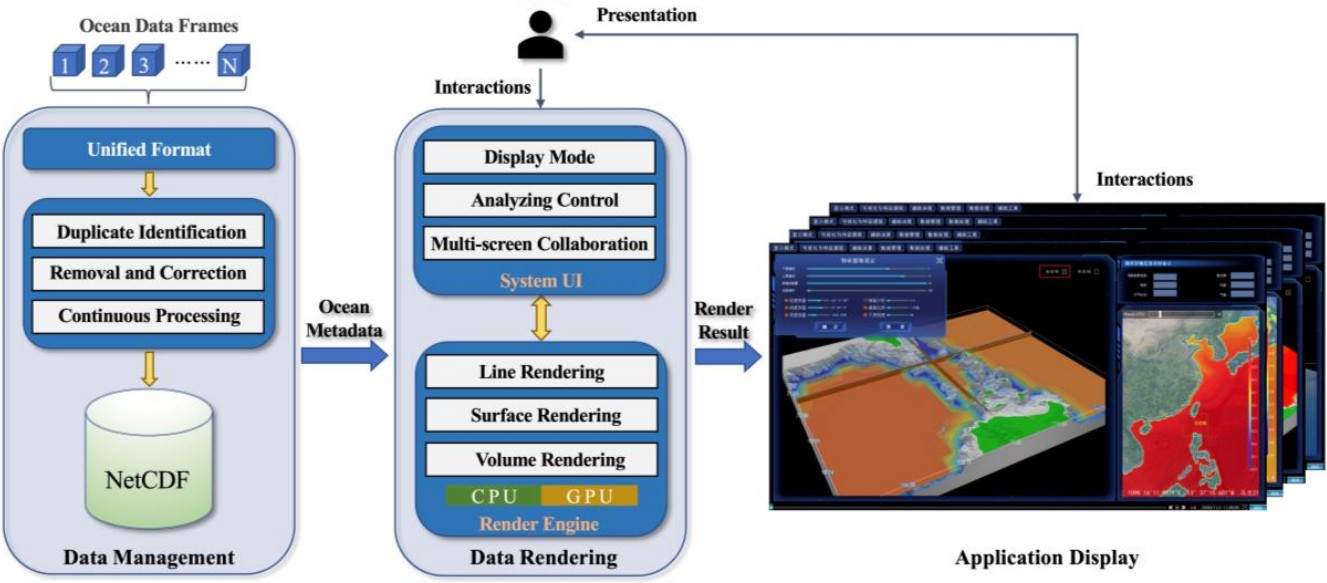

**Figure 1.** System framework.

Considering the diversity and complexity of marine environmental data, to establish a marine visualization system, one must first establish a stable and efficient marine database. The functions of the data management part mainly include: data format conversion, data preprocessing, data editing management and data display. Data management content is described in detail in Section 3.

The data rendering part is the main component of the visualization system. The data rendering module uses a multi-threaded, integrated user interface and rendering engine to complete the visual rendering and analysis of multi-dimensional marine feature data. The 3D rendering engine consists of a series of rendering modules, mainly including line rendering, surface rendering and volume rendering. These modules use Qt as the development framework and use Open GL to develop the underlying API for rendering. The rendering engine runs faster by mobilizing the GPU. The detailed optimization details of the rendering engine are described in Section 4.

The application display part is responsible for different operations of the corresponding user, such as attribute viewing, scene roaming, data analysis, and function switching. This part interacts with the data rendering part and displays the visualization results.

## 3. Data Management

The source of marine data is complex, and the amount of data is huge, so to visualize it, effective data management is necessary. The visualization system designed in this paper mainly uses NetCDF format data. NetCDF is a data format widely used in marine meteorological research. This format can efficiently process and store data in a grid. NetCDF format data are compatible with the C language programming environment, so they can be easily read by a variety of programming software [12].

In this paper, the NetCDF conversion module is designed to perform a series of preprocessing on the data, which mainly includes the following aspects.

- Convert the marine hydrological environment and meteorological environment based on NetCDF storage to the specified data format.

- Supports data preprocessing functions such as interpolation and noise reduction for the marine hydrological environment and meteorological environment stored in NetCDF.
- Support real-time interactive data editing (modification, deletion) of marine hydrological environment data stored in NetCDF, with WYSIWYG data processing function.
- Support the browsing and display of marine environmental data in the database.

The functional design framework of the data management module is shown in Figure 2.

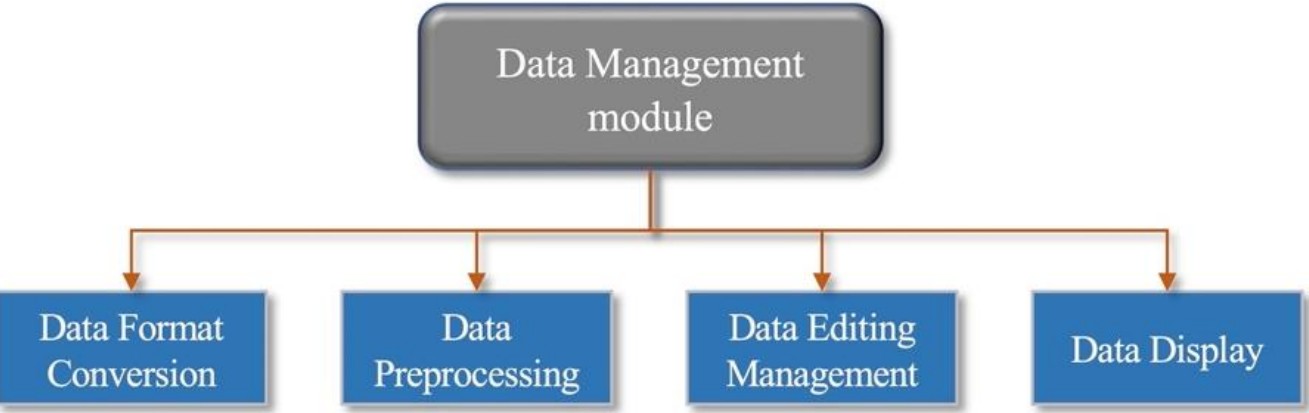

**Figure 2.** Data management module framework.

### 3.1. Data Format Conversion

In order to construct the data module, we need to provide a unified standard data format. The system uniformly converts the marine environment data format into the NetCDF format data for storage management. The data management module also supports converting the marine hydrological environment data stored based on NetCDF into a specified data format, which can be displayed and applied on the system platform.

### 3.2. Data Preprocessing

Data preprocessing can realize the functions of interpolation and noise reduction in marine hydrological environment data. Specifically, it includes: identifying and eliminating duplicate data; deleting or correcting noise data including outliers; the continuous processing of discrete marine environmental data. The following are described in turn:

(1) Identification of duplicate data. In the process of data introduction and database access, it is necessary to repeatedly identify and verify the data. The system additionally provides a window for data duplication identification, and the user can manually perform data duplication identification.

(2) Abnormal data mainly refers to mutation data and zero data. Mutational data means that the obtained data are quite different from the theoretical data; for example, the order of magnitude is quite different from that of the normal data, and the real-time data suddenly rises and falls. Zero data is the result of a malfunction in the equipment that measures and transmits. For abnormal data, it is necessary to have the operation function of deletion or correction. The abnormal data can be identified by setting the normal value range of the data in the form of a dialog box, and the identified abnormal data can be deleted or manually corrected.

(3) Continuous Processing of Discrete Data. The dynamic changes of marine elements or phenomena are continuous in space, but in fact, the data collected are often discrete. Although the total amount of data is very large, the collected data may be relatively sparse within a certain time and space range. On the one hand, it is caused by discontinuous collection methods, and on the other hand, the geographical distribution characteristics of the marine environment will also mean that the collected data are not very comprehensive. Therefore, it is necessary to interpolate or thin the data through

the spatiotemporal interpolation method to increase its spatial continuity, and to achieve a complete, smooth and efficient expression of the marine regional data.

### 3.3. Data Editing Management

The huge ocean database needs powerful data editing functions to organize and update it. The data editing management function can retrieve the corresponding data in the database with a faster response speed and realize the editing operation on it.

### 3.4. Data Display

Support the classified and hierarchical browsing display of the marine hydrological environment and meteorological environment data stored in NetCDF and support the classified and hierarchical browsing and display of the preprocessed data and the converted data.

## 4. Visualization Technology

The visualization part is the main part of the system. How to realize and combine the visualization algorithms reasonably according to the usage requirements is the focus of our research. For different marine elements and analysis requirements, this paper designed visualization algorithms such as isoline visualization, isosurface visualization, CT section visualization, volume visualization and flow vector visualization. These algorithms constitute the visualization module of the system. Through these algorithms, the visualization system designed in this paper can meet the different visualization requirements of various marine elements and has rich usage scenarios. Each visualization algorithm is described in detail below.

### 4.1. Submarine Terrain Rendering

We use the measured water depth data in the sea area and comprehensively consider elements such as contour lines and topographic feature lines (ridge lines, valley lines, contour lines, lake edge lines, etc.) to construct submarine DEM data. In order to unify the standard and format, we use data processing methods such as coordinate transformation, projection transformation, and edge processing to process the data. After that, we used principal component analysis (PCA) data fusion technology to realize the integration and complementation of multi-coordinate systems and multi-precision terrain data [13].

Since the obtained water depth data are discrete, it is necessary to construct a reasonable seabed terrain elevation model with the help of a suitable interpolation algorithm. The reconstruction of seabed topography needs to connect discrete points to form a surface according to a specific topological relationship. The surface is an approximate representation of the submarine terrain and ground objects. Its fidelity depends on the number of sampled data points and the extraction of feature points. Therefore, the sampling point is key to constructing a 3D seabed topographic surface [14,15]. The density of the existing bathymetric measurements is quite small, and it is difficult to simulate the topographic surface of the seabed. Therefore, the water depth point must be interpolated. The most commonly used method is the Kriging space interpolation method.

In this paper, the ordinary Kriging method is used to interpolate the seabed topographic data. The Kriging interpolation algorithm is suitable for regional variables with spatial correlation, and the seabed topography conforms to its random and structural characteristics [16]. The Kriging interpolation algorithm used in this paper uses the "power" variable basis function, the grid spacing is set to 1, and the "point" mode is used, the slope and anisotropy ratio are both set to 1, and the anisotropy angle is set to 0.

The process of constructing a submarine DEM using ordinary kriging interpolation is shown in Figure 3.

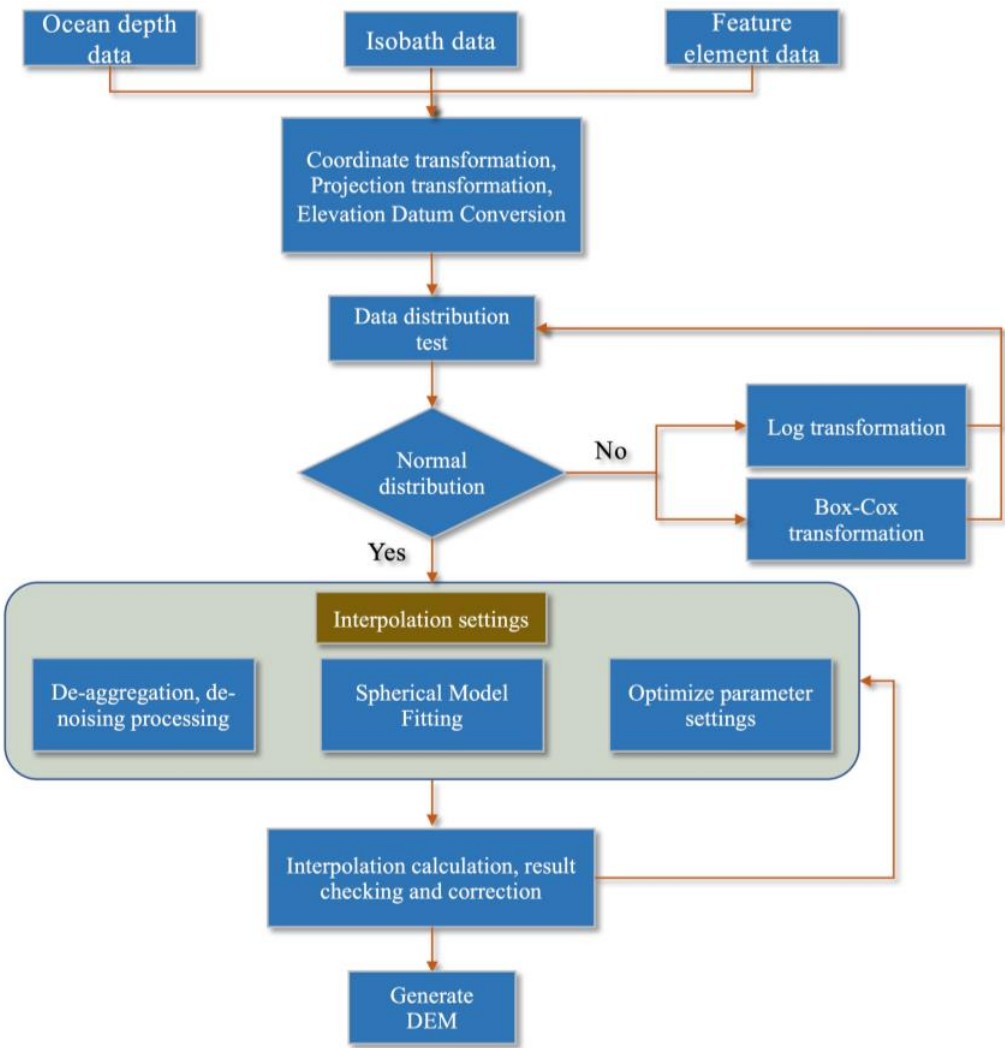

**Figure 3.** Flow chart for constructing seafloor topography DEM data.

After constructing the seabed topography data, the system uses high-resolution satellite images and DEM data to construct high-precision topography. The design framework of the 3D seabed terrain simulation is shown in Figure 4.

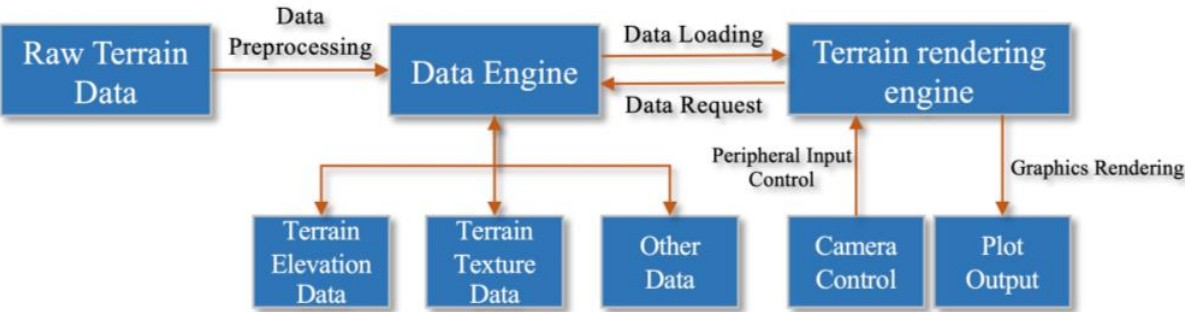

**Figure 4.** Framework for submarine 3D terrain simulation.

The satellite image data and DEM data are organized by an ellipsoid quadtree. This method of organization is easy to operate and means that the mapping between satellite imagery and DEM data is straightforward. Through the mutual mapping relationship, the satellite image will create a three-dimensional scene with the surface of the DEM, generating undulating terrain, and realizing the simulation of terrain and landforms.

When rendering, the terrain rendering engine will send a data request to the data engine and obtain the terrain data that needs to be drawn. The user realizes the roaming of the camera in the virtual earth through the input of the peripheral device, so as to obtain the terrain images in different positions and different perspectives.

The effect of the seabed terrain processing and visualization realized by the above method is shown in Figure 5.

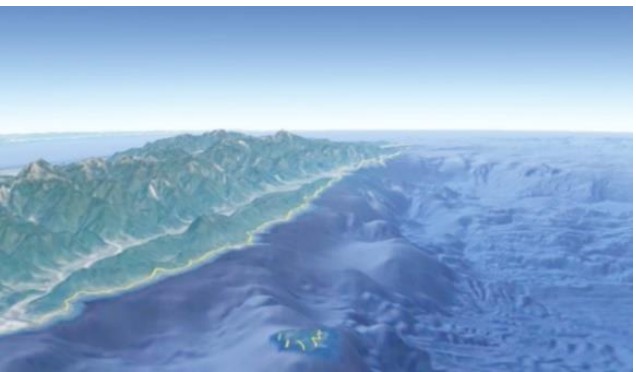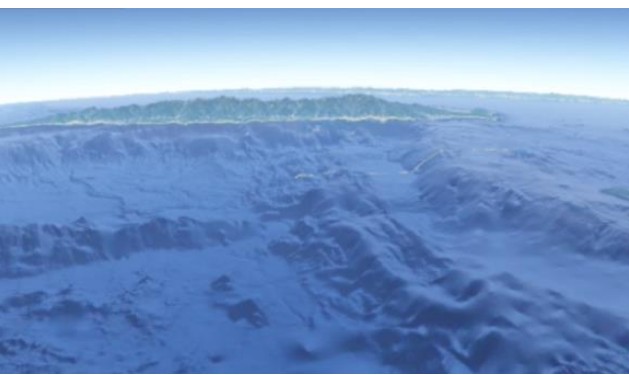

**Figure 5.** Submarine terrain visualization.

The visualization of submarine terrain provides a basic spatial framework for macroscopic and microscopic information display and various interactive analyses. On this basis, the system can effectively realize the visualization and multi-dimensional dynamic display of marine environmental data.

### 4.2. Contour Line Visualization

The contour line is a smooth curve after connecting nodes. The contour line can help customers to understand the distribution of values briefly so that it can help in the analysis of value characteristics. Contour line is widely applied in the visualization area.

#### 4.2.1. Contour Interpolation Algorithm

Suppose there are two points denoted as $d_1$ and $d_2$, and the values of them are $v_1$ and $v_2$, while the attribute value of the contour line is $v$. Calculate the value of the flag according to (1).

$$J = (v_1 - v) \times (v_2 - v) \tag{1}$$

Judge $J$ to determine whether there is an equivalent point between $d_1$ and $d_2$.

(1) If $J < 0$, it means that there is an equivalent value between $d_1$ and $d_2$. The formula for calculating the equivalent coordinates is shown in (2).

$$\begin{cases} x = d_1(x) + (v - v_1) \times \frac{d_2(x) - d_1(x)}{v_2 - v_1} \\ y = d_1(y) + (v - v_1) \times \frac{d_2(y) - d_1(y)}{v_2 - v_1} \end{cases} \tag{2}$$

(2) If $J > 0$, it means that there is no equivalent value between $d_1$ and $d_2$.

(3) If $J = 0$, it means that there is a singularity between $d_1$ and $d_2$. The existence of singular points will increase the computation time. Therefore, in order to eliminate this shortcoming without affecting the result of the contour operation, we assign the singular point a minimum value of 0.0001.

#### 4.2.2. Contour Parallel Algorithm

The grid lines of regular grid data are orthogonal to each other, and each grid cell is a rectangle. We set its four vertices as: $(x_i, y_j)$, $(x_i, y_{j+1})$, $(x_{i+1}, y_j)$, $(x_{i+1}, y_{j+1})$, and their corresponding values are: $F_{i,j}$, $F_{i,j+1}$, $F_{i+1,j}$, $F_{i+1,j+1}$.

The calculation of the intersection point between the grid unit and the contour line is mainly to find the intersection point between the edge line and the contour line of each unit. Assuming that the function varies linearly within the cell, vertex determination can be used. The grid points are divided into two states, "$-$" and "$+$", which indicate that the point is inside or outside the contour line, respectively. Let the value of the contour line be $F_t$, if $F \leq F_t$, mark it as "$-$"; if $F > F_t$, mark it as "$+$". After specifying the direction of the contour lines, the connection methods of the contour lines can be divided into the following three situations:

(1) No equivalent point. When the vertices are all "$+$" or "$-$", there is no isoline segment.
(2) The number of equivalent points is two. There are two cases: one vertex is different from the other three vertices; There are two "$+$" and two "$-$" vertices and the vertices with the same flag are on the same edge (as in Figure 6).
(3) The number of equivalent points is four. In this case, there are two ways to draw the contour line, as shown in Figure 7. The reason for such ambiguity is the existence of a saddle point in this unit. We can perform analysis of the dual linear interpolation function from inside the unit. As the linear interpolation is used at the margin of the unit, the value change of function which determines the surface of the unit is in the dual linear property, which indicates that the contour line in the unit is a hyperbolic curve rather than a straight line as in (3).

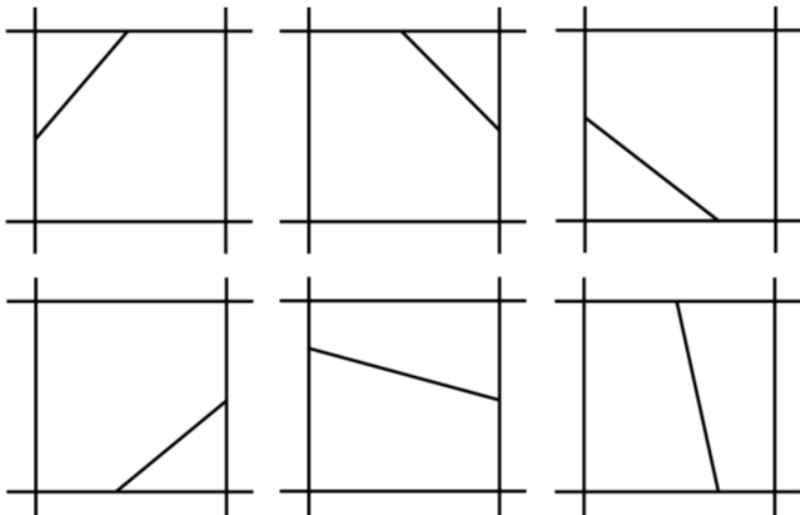

**Figure 6.** Two equivalent points.

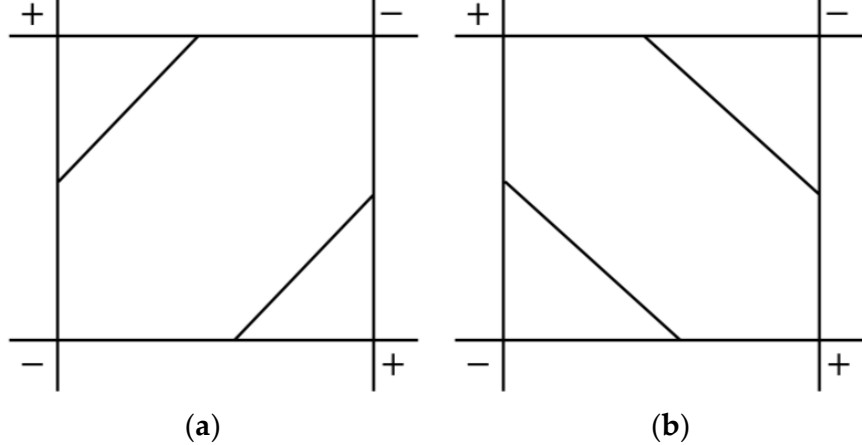

**Figure 7.** Four equivalent points. (**a**,**b**) are two possibilities, respectively.

$$F(x,\,y) = a_0 + a_1x + a_2y + a_3xy \qquad (3)$$

The ambiguous connection can be determined by the function value according to the cross point of the two asymptotic lines of the hyperbolic curve. This is because the cross point of the two asymptotic lines is always located at the same area as one of the pairs of vertices. Therefore, use the first connection approach if the cross point is "+"; use the second connection approach if the cross point is "−". In this paper, for simplification, we use a unit diagonal cross point to replace the calculation of the asymptotic lines cross point.

In this paper, the above algorithm is placed in the GeometryShader of the GPU to achieve real-time contour line extraction and visualization. We selected the monthly average salinity data of the Western Pacific in January 2022 for the visualization of contour lines, in which (a) is the sea surface salinity data, and (b) is the 200 m-deep salinity data. The effect is shown in Figure 8.

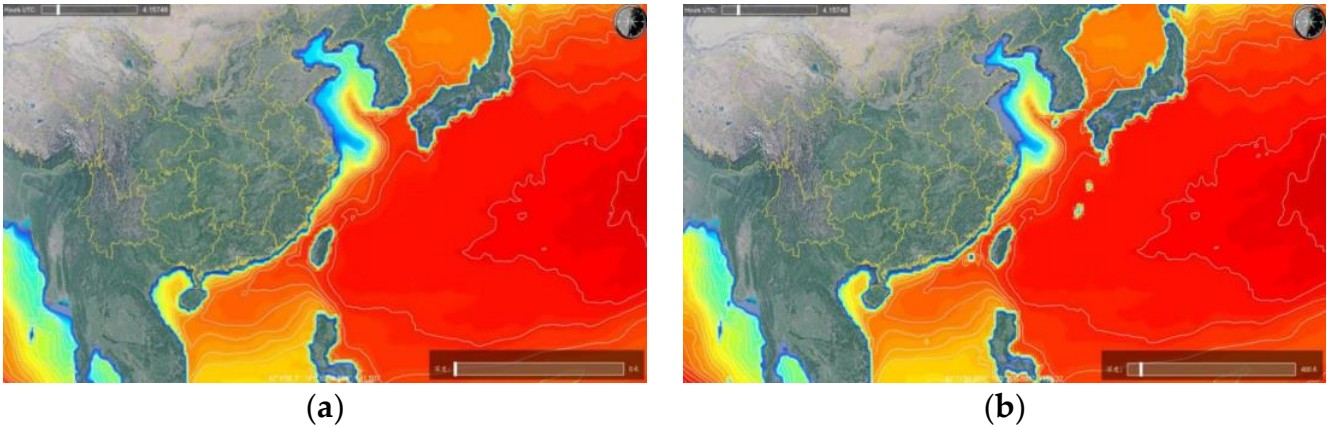

(**a**)  (**b**)

**Figure 8.** Contour line visualization. (**a**,**b**) are contour distributions at different depths.

### 4.3. Isosurface Visualization

The isosurface can be regarded as the extension of the contour line in the three-dimensional space, that is, all the point sets satisfying $f(x,y,z) = c$ in the data space [17,18]. The most commonly used isosurface extraction algorithms are Branch-On-Need Oct-tree (BONO), Near-Optimal Isosurface Extraction (NOISE) [19] and Marching Cubes (MC) [20,21]. The MC algorithm is simple in principle and easy to implement, so it is a classic algorithm for generating isosurfaces of three-dimensional data [22]. Therefore, in order to ensure the reliability of the algorithm, this paper decided to use the MC algorithm based on GPU operation. The algorithm flow is as follows.

(1)   Layer 3D discrete regular data into texture data and read it into video memory;
(2)   Scan the two layers of data in Vertex Shader and construct the voxels between the two layers one by one. Eight vertices in each voxel are taken from two adjacent layers;
(3)   Compare the value of each vertex of the voxel with the given isosurface value Ft in the Geometry Shader. Classify according to the result of the comparison, and construct the state table of the voxel;
(4)   According to the state table, find the isosurface distribution pattern corresponding to the state value in Geometry Shader, and determine the voxel boundary that will intersect the isosurface;
(5)   In the Geometry Shader, the coordinates of the intersection of the voxel boundary and the isosurface are calculated by linear interpolation. Connect these intersection points into triangles according to the corresponding mode to form the isosurface in the unit;
(6)   In the Geometry Shader, the gradient at each vertex of the voxel is obtained by the central difference method. After that, the normal vector at each vertex of the triangle is obtained by the linear interpolation method;

(7)    In the Fragment Shader, light and shadow rendering is performed according to the coordinate values and normal vectors of vertices in each triangular patch, so as to obtain high-quality and real-time isosurface images.

The visualization effect of the three-dimensional isosurface obtained by the above method is shown in Figure 9, where (a) is the parameter display during debugging, and (b) is the visualization effect of the temperature isosurface in the South China Sea (16.271° N–16.299° N, 116.688° E–116.756° E).

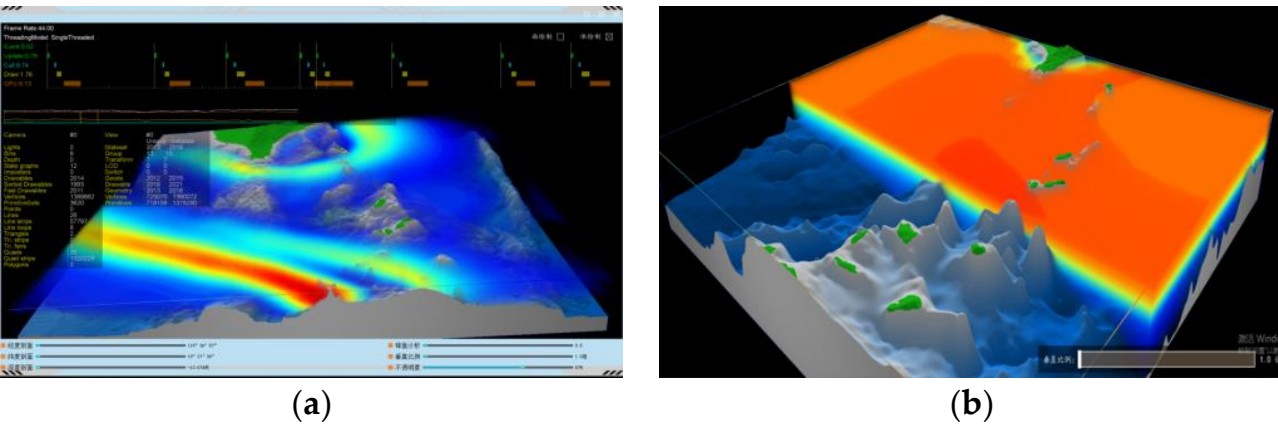

(**a**)                                                                                       (**b**)

**Figure 9.** Isosurface visualization. (**a**) is the debugging interface, (**b**) is the area display.

*4.4. Section Visualization*

By setting a certain color range, section visualization can map the entire attribute value to the specified color range to establish a corresponding relationship, so as to realize the color filling of the data field. In this way, the spatial variation of the data can be perceived from the section. In the application of 3D visualization, realizing the subdivision of the 3D space model is an important part of almost every 3D visualization system.

In section visualization, one of the most important parts is to realize the arbitrary division of three-dimensional space. This paper adopts the method of directly generating subdivision planes based on voxels. By analyzing the relationship between each voxel and the section in the 3D data set, it can directly draw and generate any section image. Therefore, the interrelationship between voxels and profiles can be translated into the interrelationships between line segments and spherical profiles. According to the three-dimensional geometric principle, in the Cartesian three-dimensional coordinate system, the spherical equation is:

$$(X - x_0)^2 + (Y - y_0)^2 + (Z - z_0)^2 - R^2 = 0 \tag{4}$$

where $(x_0, y_0, z_0)$ are the coordinates of the center of the sphere, and $R$ is the radius of the sphere.

Use $f(x, y, z)$ to represent the left-hand side expression of the spherical section equation, that is

$$f(x, y, z) = (X - x_0)^2 + (Y - y_0)^2 + (Z - z_0)^2 - R^2 \tag{5}$$

Let the spatial coordinates of a vertex in the voxel be $(x_i, y_j, z_k)$, and bring it into the spherical equation to get $f(x_i, y_j, z_k)$. When $f(x_i, y_j, z_k)$ is greater than or equal to 0, it is considered that this vertex is located outside or on the boundary of the spherical section, and its state value is set to 1. When $f(x_i, y_j, z_k)$ is less than 0, it is considered that this vertex is located inside the spherical section, and its state value is set to 0. Therefore, the state value of the voxel itself (0 or 1) determines whether the voxel can be mapped into the texture space as a texture. If the image inside the visible volume needs to be drawn after the segmentation, the visualization data with the voxel state value of 0 is mapped to the texture space, and the state value of 1 is assigned 0.

Using the above method to obtain the section visualization of the seabed topography and marine environment scalar data field is shown in Figure 10.

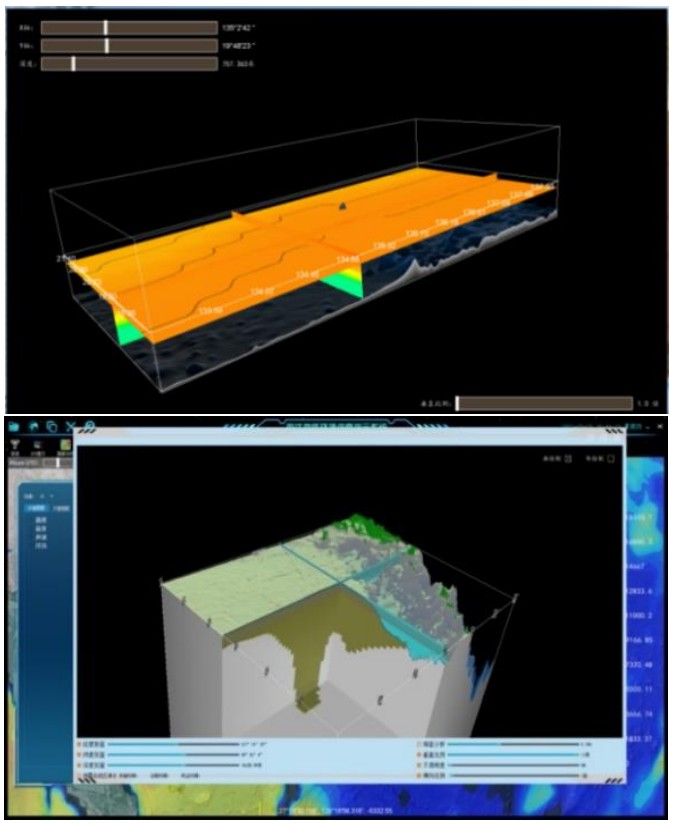
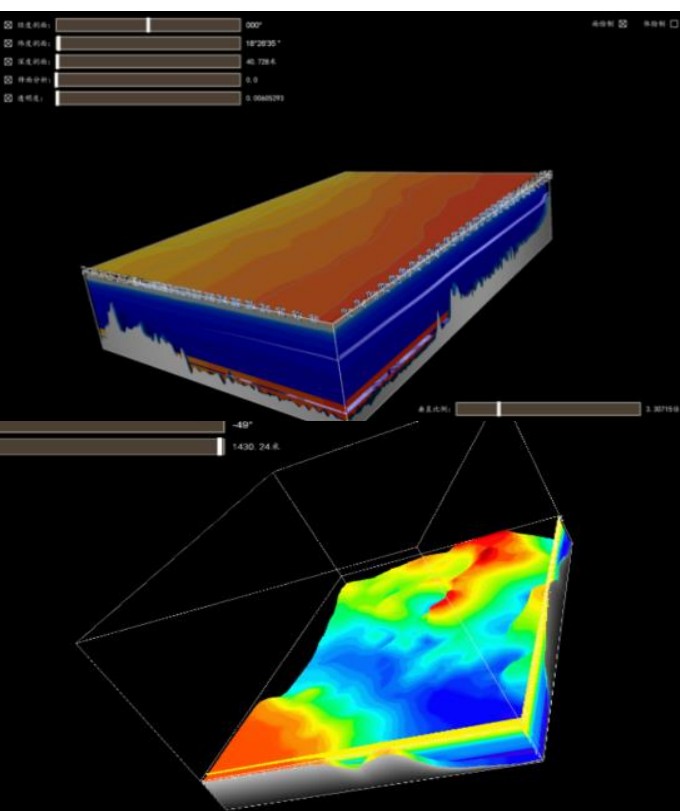

**Figure 10.** Section visualization.

### 4.5. Volume Visualization

Due to the characteristics of marine environmental elements, volume visualization can best describe the temporal and spatial changes of marine environmental elements. Volume rendering technology is a straightforward rendering method. The direct volume rendering of the data can accurately and clearly reproduce the original 3D data field, allowing users to intuitively feel the distribution and changes of data values at any position in the 3D data field.

In this paper, a GPU-based 3D data field pre-integration RayCast volume visualization method is used to realize the time evolution volume visualization of the 3D data field. Compared with other volume visualization algorithms (splatting algorithm, texture algorithm), the ray casting algorithm is considered to be in line with people's cognitive common sense [23]. In addition, the ray casting algorithm can perform parallel computing by invoking the GPU, thereby improving the operating efficiency of the visualization system. Considering the above factors, this paper decided to use the ray casting algorithm for the volume visualization of the marine environmental data [24].

The so-called pre-integrated ray-projection volume visualization refers to dividing the traditional volume, rendering numerical integration into two steps. This avoids the high sampling frequency problem of nonlinear transfer in traditional texture visualization methods [25]. The first step samples the continuous scalar field $s(x)$ in three dimensions. The second step samples the pre-integrated textures generated by the transfer functions $c(s(x))$ and $\tau(s(x))$ according to the sampling results of the first step. Since we only sampled the continuous scalar field in the first step, its frequency is not affected by the transfer functions $c(s(x))$ and $\tau(s(x))$.

In this way a single piecewise linear scalar field is constructed from the sampled values, where each linear segment volume rendering integral can be quickly obtained through a three-parameter lookup table. The three parameters of the search table are the pre-integration scalar value $S_f = s(x(id))$, the post-scalar value $S_b = s(x((i+1)d))$ and the integral segment length $d$. The pre-integrated $s(x))$ piecewise linear sampling principle is shown in Figure 11.

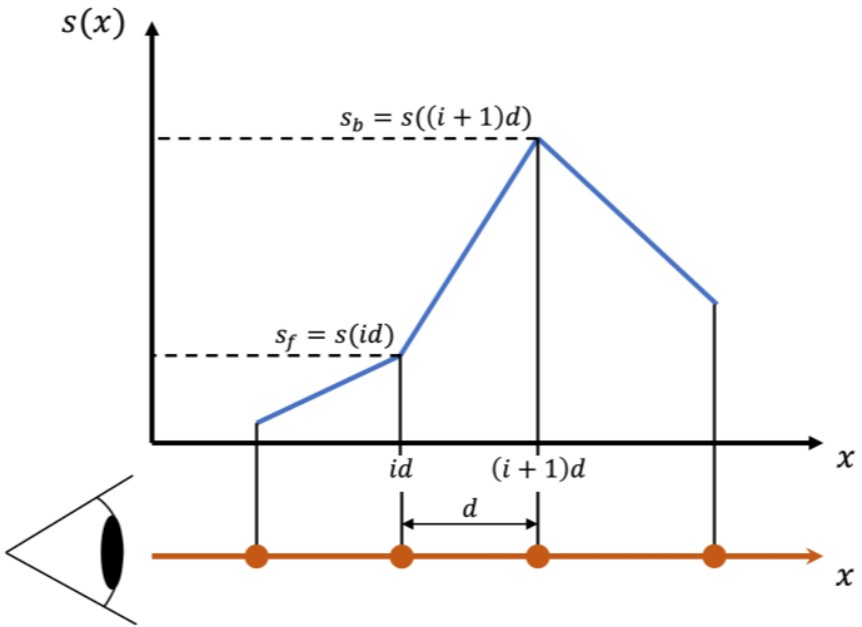

**Figure 11.** Pre-integration piecewise linear sampling principle.

Then, the opacity $a_i$ of the $i$-th segment can be approximately expressed as:

$$a_i = 1 - \exp\left(-\int_{id}^{(i+1)d} \tau(s(x(\lambda)))d\lambda\right) \approx 1 - \exp\left(-\int_0^1 \tau\left((1-\omega)s_f + \omega s_b\right)dd\omega\right) \quad (6)$$

It can be seen from the above formula that $a_i$ is a function of $s_f$, $s_b$ and $d$ (when $d$ is equal, $a_i$ is a function of $s_f$ and $s_b$). The color $\tilde{c}$ of the corresponding $i$-th segment is a function of $s_f$, $s_b$, and $d$:

$$\tilde{c}_i \approx \int_0^1 \tilde{c}\left((1-\omega)s_f + s_b\right) \times \exp\left(-\int_0^\omega \tau\left((1-\omega')s_f + \omega' s_b\right)dd\omega'\right)dd\omega' \quad (7)$$

At this time, the pre-integrated volume rendering formula is:

$$I = \sum_{i=0}^n \tilde{c}_i \prod_{j=0}^{i-1}(1 - a_j) \quad (8)$$

where $I$ represents the accumulated light energy, $\tilde{c}_i$ and $a_j$ in the formula can be calculated in advance, respectively. The associated color during the calculation can be approximated by the following equation:

$$\tilde{c}_i \approx \int_0^1 \tau\left((1-\omega)s_f + s_b\right)c\left((1-\omega)s_f + s_b\right) \times \exp\left(-\int_0^\omega \tau\left((1-\omega')s_f + \omega' s_b\right)dd\omega'\right)dd\omega \quad (9)$$

It can be seen from the above description that in the pre-integrated volume visualization method, regardless of whether the transfer function is applied or not, the pre-computed color is always associated. Since the pre-integral volume rendering first samples the continuous scalar field $s(x)$ without having to increase the sampling rate like the nonlinear

transfer function, it effectively avoids the high sampling rate problem of the nonlinear transfer function in 3D data field volume visualization.

The steps of the GPU implementation method of pre-integrated volume visualization are shown in Figure 12.

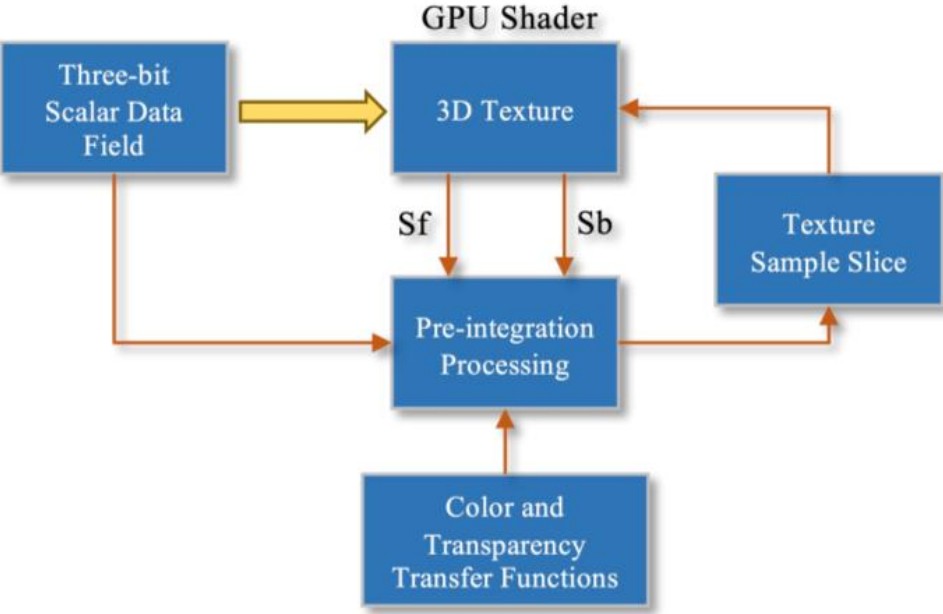

**Figure 12.** Pre-integrated volume visualization implementation steps.

As shown in the figure above, in the initialization stage, the 3D scalar field is imported into the GPU's video memory as a texture. Additionally, it computes a pre-integrated texture based on the set color and transparency transfer functions and the extent of the scalar field. The result is also imported into the GPU's video memory. Second, in the draw-through, the 3D texture is sampled with texture slices. Obtain $S_f$ and $S_b$ in the pixel shader of the GPU and apply $S_f$ and $S_b$ as coordinates to sample the pre-integrated texture. Use the sampled result as the final color value of the texture slice. Finally, the drawn texture slices are mixed with transparency and color in order from back to front to obtain the final image.

In order to achieve a better rendering effect, the texture filtering type in this article uses the linear type. For invalid data points, set their transparency to 0 by adding a judgment flag. In order to achieve dynamic rendering, the data in the three-dimensional space needs to be able to be updated in real time.

Based on this method and idea, the system designed in this paper can realize the rapid visualization and related solutions of ocean physical fields. The real-time 3D dynamic visualization of various marine mesoscale phenomena can also be realized based on the measured marine environmental data. The system deeply mines the spatial distribution and dynamic evolution characteristics of marine environmental data through volume visualization and realizes the comprehensive analysis and information extraction of key characteristics of the marine environment.

The volume visualization effect of the system is shown in Figure 13.

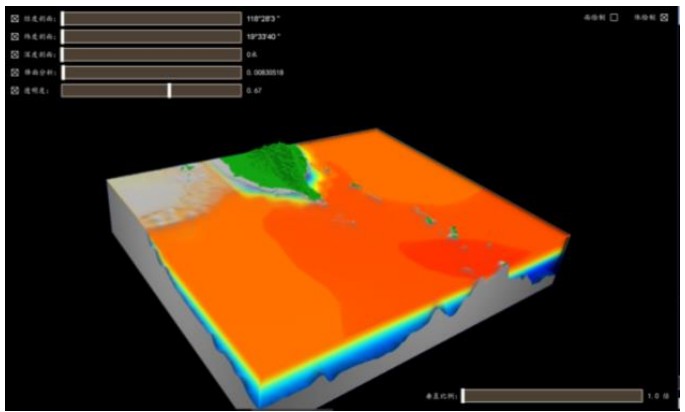 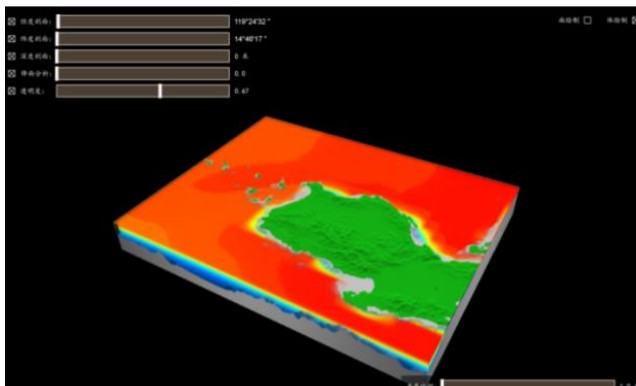

**Figure 13.** Volume visualization.

### 4.6. Flow Vector Visualization

In this paper, the particle system algorithm is used to visualize the flow vector. The particle system algorithm uses particle sets with certain properties, random distribution and continuous motion to simulate irregularly moving objects, which has a strong sense of reality. Due to the characteristics of particle systems, it is widely used in the visualization of flow field vectors [26,27].

The main problem of the real-time rendering of ocean flow field lies in the speed of the operation of the flow vector data and the improvement of the image rendering speed. For time-varying ocean current dynamic nodes, each node takes a long time to compute and render.

In order to solve this problem, this paper uses the CUDA model to realize the parallel operation of CPU and GPU [28,29]. A Computation Node is added in the scene as the vessel of My Module and My Resource. Parallel computing is conducted with the module. Vertex array resources are mapped to the device memory through the 'map' function. 'cudaMemcpy' is invoked in 'Launch' and ocean current data are transferred to GPU for parallel computing through 'cudaMemcpy HostToDevice'. The mapped vertex array is updated. The rendering of the ocean current visualization is shown in Figure 14.

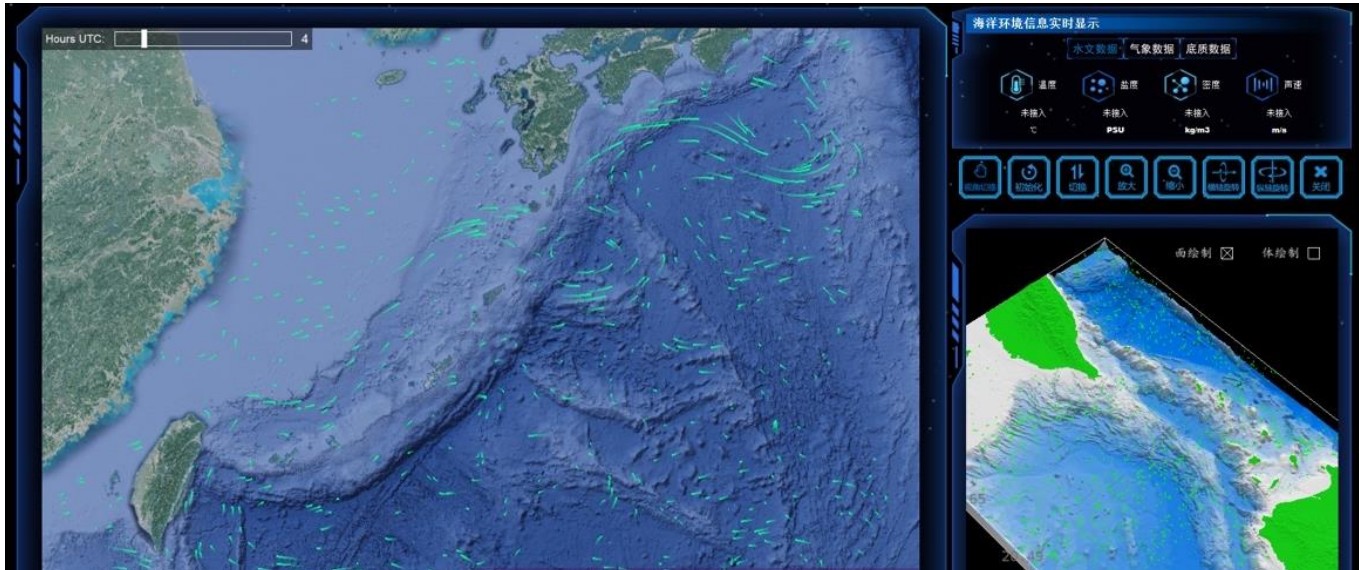

**Figure 14.** Ocean current visualization.

Table 1 compares the rendering efficiency of the traditional method and the GPU parallel method. The results show that CUDA can guarantee the visualization effect output of about a 50 frames/s frame rate when the vector magnitude is $1 \times 10^5$, which is much more effective than the traditional method.

**Table 1.** Rendering Results Comparison.

| Rendering Method | Data Volume (Ocean Current) | Frame Rate |
| --- | --- | --- |
| Traditional | $1 \times 10^5$ | 2.5 frames/s |
| CUDA | $1 \times 10^5$ | 50 frames/s |

## 5. Software Function Realization

This article uses Visual Studio 2013 and Qt5.5 environment to complete the development of the marine environment visualization software. The visualization algorithm developed in this paper can quickly respond to operations. When the visualization module is not lower than the given operating environment and the data field resolution is lower than $256 \times 256 \times 64$, the volume visualization resolution can reach not less than 24 frames/s. The screen resolution can reach not less than $1920 \times 1080$.

In order to meet the analysis needs of scientific researchers for marine environmental data, based on the development of visualization algorithms, we have also developed a series of data analysis functions, including: (1) point query; (2) local analysis; (3) multi-screen collaboration.

Next, we will introduce the above data analysis functions in further detail.

### 5.1. Point Query and Line Query

According to the different characteristics of marine environmental elements, this function realizes the query of information points in the time domain, area and air domain. The elements to be investigated are displayed in the form of data and curves, and the spatial characteristics and changing trends of multi-dimensional dynamic marine environmental data are comprehensively detailed and accurately displayed. As shown in Figure 15a, the red box in the point query window represents the data corresponding to different depths, and the green box shows the data change curve. The mouse wheel can zoom in and out of the change curve, and the mouse drag can control the curve position. As shown in Figure 15b, the starting and ending position information at the bottom of the line query window is the latitude and longitude information of the starting and ending positions of the connecting line in the line query. The left side of the window displays the queried environmental profile information, and the right side displays the data curve.

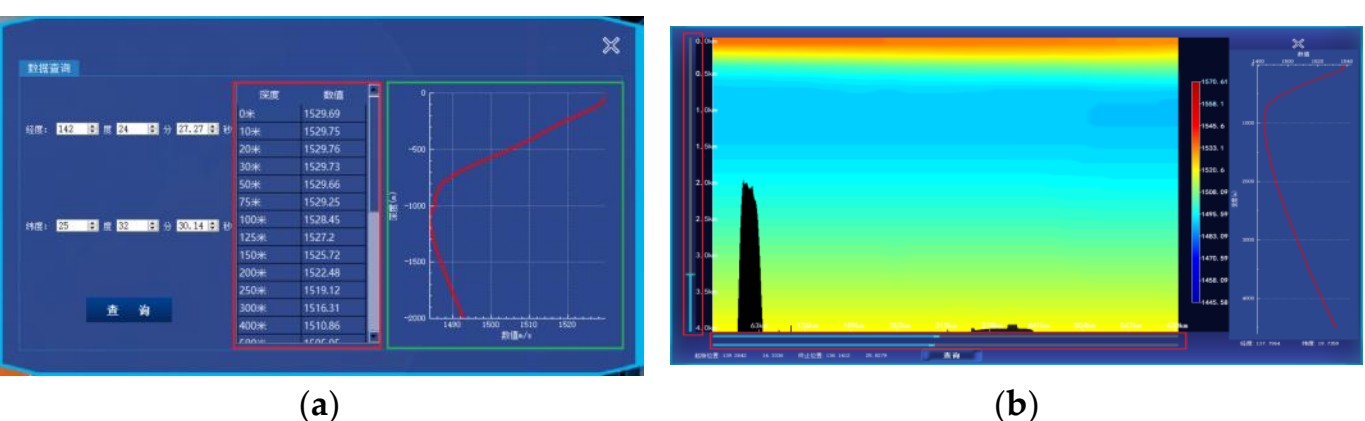

**Figure 15.** Point query (**a**) and line query (**b**) renderings.

### 5.2. Local Analysis

In order to facilitate users to perform local analysis on the regional marine environment, this paper proposed a local analysis function. This function supports the contour visualization, section visualization, frontal analysis, transparency adjustment and other functions of the selected sea area. The local analysis function can be selected between the surface drawing and volume drawing modes, and the default drawing method is surface drawing. In the volume rendering mode, the data are displayed semi-transparently. The effect of the local analysis is shown in Figure 16.

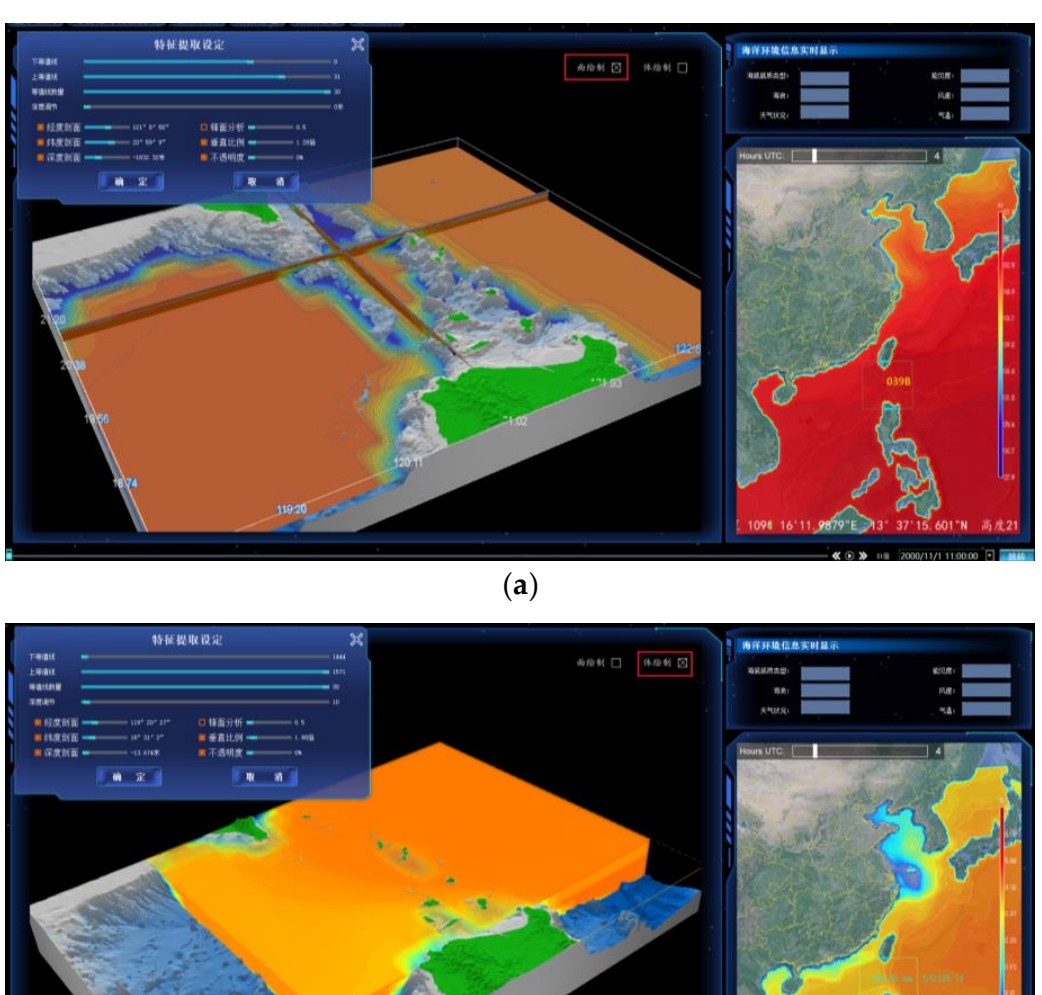

(**a**)

(**b**)

**Figure 16.** Local analysis renderings ((**a**) is surface rendering, (**b**) is volume rendering).

### 5.3. Multi-Screen Collaboration

The visualization system designed in this paper has two windows: the main screen and the sub-screen. The main screen defaults to display the top-view perspective, and the sub-screen defaults to the third-person perspective. The specific display situation is shown in Figure 17. The software has designed a screen switching button, which can exchange the display content of the two screens to meet the needs of users in various scenarios. By combining visual images of different perspectives and scales, users can compare and analyze the local characteristics of the area while observing the changes in

the distribution of the overall environmental elements. The introduction of multi-screen windows improves the user experience, meets the needs of multi-screen collaboration and improves the efficiency of data analysis.

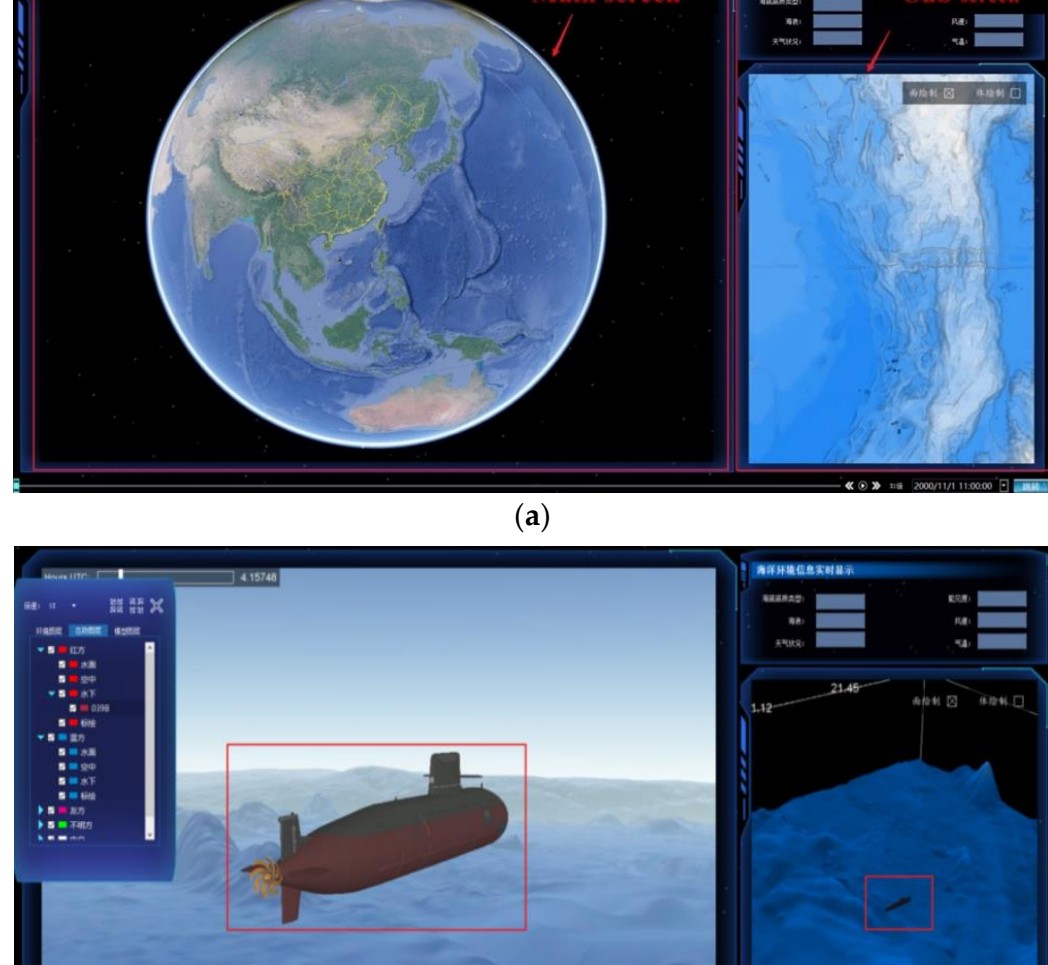

(**a**)

(**b**)

**Figure 17.** Multi-screen display effect. (**a**) is the overview of the multi-screen display, (**b**) is the multi-screen display in the submarine scene.

## 5.4. System Performance Test

This paper conducts a performance test on the system to verify whether it meets the usage requirements. The operating environment for the performance testing is shown in Table 2.

**Table 2.** Performance test running environment.

| | type | Core |
|---|---|---|
| Processor | CPU | 9th Generation Core i7 |
| | CPU frequency | 2GHz |
| Memory | Number of memory slots | 2(DDR4) |
| | Memory Capacity | 16GBytes |
| Display performance | Display chip | Nvidia 3080 |
| | Video memory | 6GBytes |
| storage | Internal hard drive number | 2 × 3.5 inches |
| | SATA hard drive capacity | 1TBytes |
| | SSD hard drive capacity | 256GBytes |

The test content mainly includes: the data extraction speed, scalar field data rendering speed, vector field data rendering speed and running frame rate. The scalar field data of temperature, salinity, density, sound speed, etc. are from the National Marine Data Center (NMDC) of China, Tainjin, China. The flow vector data are the analysis data from the National Oceanic and Atmospheric Administration (NOAA) HYCOM (Hybrid Coordinate Ocean Model), Miami, FL, USA.

The system conducts experiments in three different areas and records information such as the data extraction speed, visualization rendering speed and system running frame rate. Ten individual experiments were performed in each area, and the results were averaged.

The experimental area 1 is the South China Sea area (6.500° N–6.600° N, 131.5° E–131.6° E); the experimental area 2 is the Gulf of Mexico area (26.500° N–26.600° N, 93.5° W–93.6° W); the experimental area 3 is the Red Sea area (23.3° N–23.4° N, 36.9° E–37.0° E). The data field resolution is 256 × 256 × 64, the screen resolution is 1920 × 1080. The results of the experiment are shown in Table 3.

**Table 3.** Data extraction speed test results.

| Extraction Type | Env1 | Env2 | Env3 |
|---|---|---|---|
| Point | 16.2 ms | 15.7 ms | 16.8 ms |
| Contour line | 27.4 ms | 28.2 ms | 27.9 ms |
| Scalar Field Rendering | 235 ms | 257 ms | 248 ms |
| Vector field rendering | 135 ms | 142 ms | 138 ms |
| running frame rate | 50 frames/s | 51 frames/s | 50 frames/s |

It can be seen from the results that the performance of the system is good, which can meet the needs of users for the visualization system in specific scenarios.

## 6. Conclusions

The original intention of the visualization system is to meet the data analysis needs of users, so it must have the characteristics of simplicity, reliability and versatility. Based on these points, the system designed in this paper fully considers the spatial and temporal characteristics of marine data and the needs of users and constructs a marine environmental data visualization system with real-time, dynamic and interactive characteristics. The system can realize the geometric analysis and dynamic rendering of large-scale marine environmental data.

In order to meet the data analysis needs of users, the system design implements visualization algorithms such as terrain rendering, contour line visualization, isosurface visualization, section visualization, volume visualization and flow vector visualization. These algorithms address the visualization needs of ocean elements including temperature, salinity, density, speed of sound and ocean currents. Terrain rendering can provide a

sufficiently reliable and realistic terrain background to the visualization system. The visualization of the contour line and isosurface can provide an effective reference for analyzing data trends. Section visualization can complete the tracking analysis of section changes. Volume visualization can effectively display ocean data and ensure that users can obtain the desired visualization angle. Flow vector visualization complements the system's vector visualization capabilities, enabling the system to achieve a full range of ocean data visualization.

In order to improve the scientific research and analysis capabilities of the system, we designed data feature extraction and analysis functions, such as point query, line query and local analysis. These functions mean that the visual analysis of data is more concise and intuitive, and effectively improve the data analysis capability of the system. In addition, the system innovatively introduces the structure of the main and sub-screens, which meets the user's needs for multi-screen collaborative data processing. The introduction of this function enables users to observe the marine environment in the same area from different perspectives at the same time, which improves the analysis efficiency. In order to improve the rendering speed of visualized images, we designed it to use the GPU platform for parallel computing.

As a data analysis method, data visualization can effectively adapt to the inherent characteristics of marine data, so it will become the main means of marine data analysis. With the dual development of graphics cards and algorithms, the efficiency of data visualization will inevitably be significantly improved, and the current problems that limit data visualization will be solved. Therefore, we believe that in the foreseeable future, data visualization will become one of the most important means of marine data analysis.

**Author Contributions:** Conceptualization, Jun Fu and Teng Lv; methodology, Teng Lv; validation, Jun Fu, Teng Lv and Bao Li; formal analysis, Jun Fu; data curation, Bao Li; writing—original draft preparation, Teng Lv; writing—review and editing, Jun Fu; visualization, Jun Fu; supervision, Jun Fu; project administration, Teng Lv. All authors have read and agreed to the published version of the manuscript.

**Funding:** This research received no external funding.

**Institutional Review Board Statement:** Not applicable.

**Informed Consent Statement:** Not applicable.

**Data Availability Statement:** Not applicable.

**Conflicts of Interest:** The authors declare no conflict of interest.

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
