# Peer review of "Design and Application of Multi-Dimensional Visualization System for Large-Scale Ocean Data"

_ijgi, doi:10.3390/ijgi11090491_

Round 1

Reviewer 1 Report

The number of references is few and there is a lack of literature from the last three years.

This paper only shows the visualization results with graphical form, and there is a lack of comparative analysis with existing methods regarding the rendering effect, real-time and visualization effect.

In line212, the phrase 'contour lines' repeats twice.

Reviewer 2 Report

I am somewhat confused about the usefulness of this paper.  This looks like a visualization framework geared to marine sciences, but I fail to understand how it improves on the capabilities of standard visualization applications such as ParaView, VisIt, Ensight, yt, etc.  I am most familiar with ParaView so I will speak from that perspective.  ParaView has netCDF readers, GPU parallelism, GIS capability, etc.  ParaView has all the analysis and visualization functionality (line, surface, volume rendering, etc.) described in this paper, with similar ability to layer multiple data sources in a pipeline.  ParaView does interactive visualization, handles sparse and missing data, etc.  No, I don’t work for ParaView and this paragraph applies equally to other visualization applications.

Is it that the authors don’t have access to common viz tools?  If there is a valid reason that they need to build an in-house version, then this paper makes sense.  But that should be clearly communicated. 

Otherwise, the paper needs to clearly state the advantages of its framework over other off-the-shelf visualization tools.  Is this system faster?  There are no overall performance numbers in the paper so we don’t know.  The only table (with a bad title, Line 500) shows GPU vs CPU performance and no one is surprised by GPU improvement.  Is this system more intuitive or easier for a domain scientist to use?  Can the user get to science answers faster?  There are neither user evaluations nor case studies in the paper.  Were users involved in the development of the framework design requirements?  Lines 82-83 make it sound like the authors decided on the requirements.

There are whole bodies of research on contour visualization algorithms, volume rendering, flow visualization, etc. yet very little background research is cited.  The paper needs discussion on why one particular algorithm was chosen versus some other possibilities.  These is no streamline visualization, as far as I can tell.  It seems like marine data would benefit tremendously from streamline visualization. 

Given decades of research continuously proving the non-uniformity of the rainbow color map, it was very depressing to see it used in the visualization system.  Even worse, Lines 451 & 462 suggest that the non-uniform color map is baked into the system.  Am I understanding that correctly?  If the pre-computed color is already associated with the data, then a non-linear transformation is guaranteed, making it impossible for the user to post hoc apply a different color map to the underlying data and calling into question any analysis based on the transformed data.  Are there other color map options?  

Table 1 – Needs the units of the data and this needs an actual title.  All figures need more informative captions. 

Reviewer 3 Report

Authors provide description of some data visualization system for multi-dimensional data. Data for visualization are declared as large-scale ocean data. However, all demonstrated examples of visualization do not refer to any specific ocean data except the ocean depth data (bottom topography). The system description looks oriented mainly on visualization system developers, not end-users, but in such case it lacks details and comparison with other visualization methods and systems. In some cases (part 4.5 as example) a lot of formulas are given which are hard to understand in contents of paper. For the potential end-users paper is mainly declarative, without clear examples of applications and system abilities in data presentation and analysis. I recommend  to reconsider the publication after major revision. My comments on some unclear points follows.

1.Abstract. Line 11,16 and all through the text: language: "this paper develops...", line 15: "this" paper designs...", line 20: "system developed in this paper", etc - please check the language and grammar through the paper.

2. Introduction. Authors mention existing visualization systems used for the ocean data presentation and introduce their approach. On my opinion description miss information on specialized visualization packages often used by oceanographers like Ocean Data View, Ferret, GMT, GrADS etc.  How new technologies (like usage of GPU) improve standard ocean data visualization tasks?

3. Multiple cases of duplicate text (copy/paste) need correction: lines 147-150 = lines 159-162;  lines 259-261 = lines 263-265

4. Additional references are necessary in some cases, like line 215: what are under "data fusion technology"?

5.Section 4.2 - details behind contour lines constructions are given, but there are no example with real ocean data.

6.Duplicate content lines 293-294 and lines 315-317.

7. Line 345: "VetexShader"? Line 348, "GeometryShander"?

8. Figure 9 - what is demonstrated?

9. Lines 415 - 418: very long single sentence.

10. Part 4.5, "Volume visualization" include many technical details, which are out of my understanding. What is difference with other methods? Why do not give an examples of different approaches? Why given description is named as "analysis" (lines 450-455)?

11. There are two Figures 15: line 543 and line 557.

12. Authors mention "flow vector visualization" (line 568) but there are no examples of such visualization.

13. References. Missing are DOI for majority of references, even when such exist and can be found. References [4] need details.

Reviewer 4 Report

The reviewed paper is of a relatively good level. In general, my comments mainly concern the structuring of the text. I believe it is within the power of the authors to edit the text and add the missing concrete information. In some parts of the paper, the quality of the text would benefit from a more detailed analysis of literature dealing with geovisualisation or digital (interactive) cartography. See detailed comments below:

-       I recommend that the authors revise the text concerning repetitive information. In section no. 3 (Data management), the same sentences are repeated three times (page 3, rows 120-123; page 4, rows 147-150, and page 4, rows 159-162).

-       Sections 3.1 and 3.2 contain, in my opinion, a minimum of factual information. I recommend that the authors significantly shorten them or, on the contrary, add more specific data here (e.g., names and citations of used procedures, algorithms, etc.)

-       Some phrases do not belong in a scientific text (e.g., NetCDF has an excellent C language interface).

-       Section 4.1 describes, among other things, kriging interpolation. However, it has several parameters and possible settings. There is no mention of them here.

-       Sections 4.2 and 4.5 contain equations. It would be convenient to format them according to the templates provided by MDPI and, above all, to explain the variables used.

-       Section 5, among other things, mentions testing the created software. The claim is that "the visualization algorithm developed in this paper can quickly respond to operations". It is a general statement and not confirmed by facts. Authors should demonstrate this with the results of some performance testing.

-       The concepts described in 5.1 to 5.3 fit into the issue of geovisualisation (or exploratory cartography). The mentioned functionality (details on demand, multiple views, spatial queries, etc.) was already described in studies thirty years ago. See the work of Monmonier, Di Bias, Andrienko or Roth. Currently, scientific studies are also devoted to applying these concepts within 3D visualization or virtual reality. The authors could also compare their results with the results of similar studies.

Round 2

Reviewer 1 Report

additional experiments needed

Reviewer 2 Report

I simply don't see this paper as a major contribution to the field.  However, the authors were responsive to most of the comments.  I assume a user evaluation or heuristic review by the user is out of the question due to the nature of the client.

Reviewer 3 Report

After revision the goals of paper become clear compared to the original version. I consider it possible to publish paper after minor corrections = please once more check the language of paper.

Reviewer 4 Report

All my comments were satisfactorily incorporated into the text of the paper and the edits were also commented on in the cover letter.